# HydraSum: Disentangling Stylistic Features in Text Summarization using Multi-Decoder Models

## Abstract

Existing abstractive summarization models lack explicit control mechanisms that would allow users to influence the stylistic features of the model outputs. This results in generating generic summaries that do not cater to the users needs or preferences. To address this issue we introduce HydraSum, a new summarization architecture that extends the single decoder framework of current models, e.g. BART, to a mixture-of-experts version consisting of multiple decoders. Our proposed model encourages each expert, i.e. decoder, to learn and generate stylistically-distinct summaries along dimensions such as abstractiveness, length, specificity, and others. At each time step, HydraSum employs a gating mechanism that decides the contribution of each individual decoder to the next token's output probability distribution. Through experiments on three summarization datasets (CNN, Newsroom, XSum), we demonstrate that this gating mechanism automatically learns to assign contrasting summary styles to different HydraSum decoders under the standard training objective without the need for additional supervision. We further show that a guided version of the training process can explicitly govern which summary style is partitioned between decoders, e.g. high abstractiveness vs. low abstractiveness or high specificity vs. low specificity, and also increase the stylistic-difference between individual decoders. Finally, our experiments demonstrate that our decoder framework is highly flexible: during inference, we can sample from individual decoders or mixtures of different subsets of the decoders to yield a diverse set of summaries and enforce single- and multi-style control over summary generation.[1]

## 1 Introduction

Abstractive summarization (Rush et al., 2015; See et al., 2017) involves a combination series of generation decisions, such as what content to directly copy from the input document and what content to paraphrase, the level of specificity vs generality, length, readability, etc. of generated summaries. Current summarization systems (Lewis et al., 2020; Zhang et al., 2020) implicitly encode these decisions in their parameters, but provide no mechanism for end users to control generation along these different axes, or to obtain a diverse set of summaries for a given input. Commonly used sampling methods such as beam search, top-k decoding (Fan et al., 2018b) or diverse decoding (Vijayakumar et al., 2018) tend to output stylistically similar summaries, and cannot be queried for multiple diverse summaries satisfying a target set of features or styles.

In this paper, we propose a new summarization architecture - HydraSum that disentangles these different stylistic decisions made by abstractive summarization models from the models weights into an explicit model component. Our model contains a single transformer-based encoder to encode the input document and a mixture-of-experts framework with multiple decoders for summary generation. At each time step of the generation phase, the next token's probability distribution is computed by combining the output probabilities obtained from each individual decoder. This allows the model to distribute the diverse stylistic and lexical features encountered in the training data, even those within the same reference summary, across the parameters of separate decoders. As an example, consider a 2-decoder scenario in which one decoder learns to only copy phrases or words

---

[1]We will share all relevant code, data and model checkpoints to support further research.

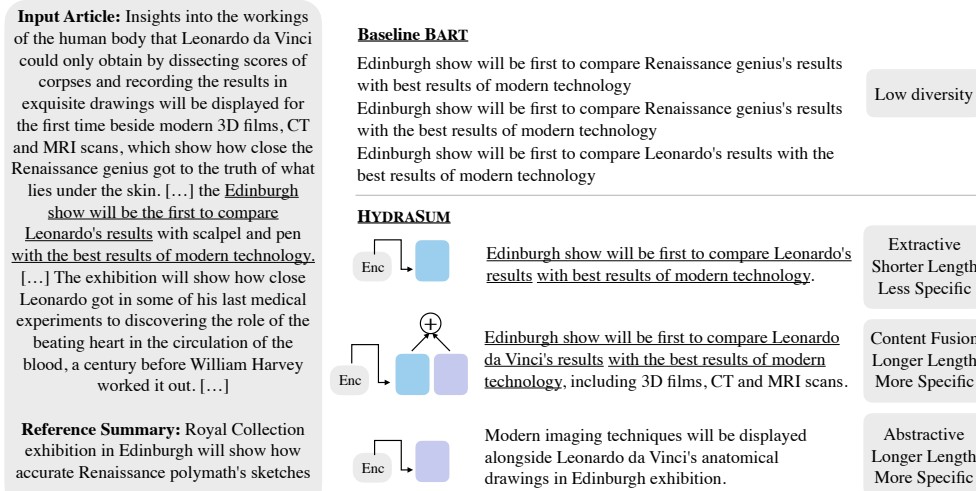

Figure 1: Examples of generated summaries for a NEWSROOM article using both BART and a 2-decoder HYDRASUM model. Longer copied sequences (denoting extractive behavior) are underlined. For HYDRASUM, summaries from different mixtures of decoders differ in degree of abstractiveness, specificity, and length.

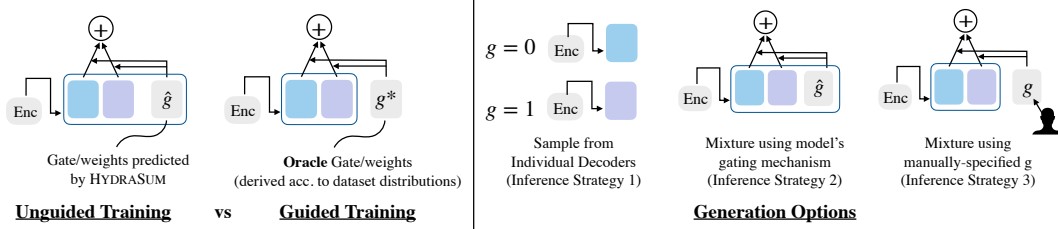

Figure 2: HYDRASUM's training and generation workflow for a 2-decoder version. We consider two training settings, namely guided and unguided, which differ in how the gate weights used to combine different decoders are obtained. During generation, summaries can be sampled from individual decoders or their mixture.

from the input document, while the second decoder only learns paraphrasing and syntactic transformations. While individual decoders cannot cover the range of stylistic variations in the dataset, a weighted combination or mixture of the two decoders can be used to model the summarization dataset. In Figure 1, we show an example of the actual style partitioning by a 2-decoder version of HYDRASUM across individual decoders. Compared to the baseline, HYDRASUM generates a more stylistically distinct set of summaries by varying the degree of abstractiveness and summary length, or including details such as *3D films, CT and MRI scans* to vary specificity.

We train and evaluate our proposed model under two settings (see Figure 2): the ***unguided*** setting in which we do not explicitly control this partitioning of the summary features and the ***guided*** setting where different decoders are trained to learn contrasting summary styles along one specific feature, e.g. low abstractiveness vs high abstractiveness. Our experiments on three summarization datasets (CNN, NEWSROOM, XSUM) shows that the proposed model exhibits significantly better stylistic-diversity and improvement in Top-K quality compared to baseline models. Moreover, we demonstrate that the flexibility of HYDRASUM's model architecture allows us reliably enforce single-style control by sampling from any combination of available decoders. In fact, these decoders can even correspond to separate HYDRASUM models and orthogonal features to provide multi-style control over summary generation.

## 2 METHODOLOGY

Current state-of-the-art summarization models (e.g. BART, PEGASUS) leverage transformer-based encoder-decoder architectures. Similarly to those models, HYDRASUM consists of an encoder network that accepts the document $x$ as input. The decoder network, however, is modified to incorporate $k(> 1)$ decoders, $\phi_1, \phi_2, ...\phi_k$, as depicted in Figure3. At time step $i$, each decoder outputs a probability distribution $P_{\phi_k}(y_i|x, y_{<i})$ over the vocabulary, corresponding to the next-token prob-

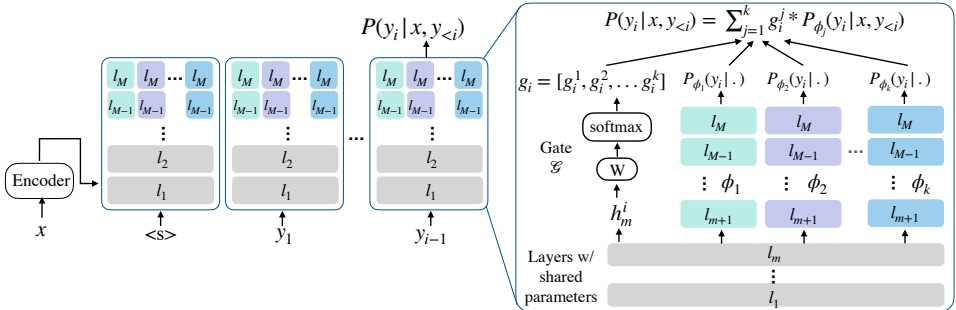

Figure 3: Our proposed HYDRASUM architecture. The decoder network incorporates multiple decoders and a gating mechanism is used to combine their output probabilities in a mixture-of-experts formulation.

abilities. The final output probability $P(y_i|x, y_{<i})$ is computed as a mixture of these $k$ probability distributions, with the mixing coefficients predicted by a gating mechanism $\mathcal{G}$.

**Multi-Decoder Architecture**: Let $M$ be the total number of decoder blocks in a single decoder: e.g. $M = 12$ for the commonly used BART-LARGE. In our model, the parameters of the $m(< M)$ bottom layers are shared between the $k$ decoders. This reduces the number of extra parameters introduced into the model architecture. The top $M - m$ layers of the different decoders are independently trained. The right block of Figure 3 shows a detailed view of the multi-decoder architecture at a single time step $i$.

**Gating Mechanism**: A gating mechanism $\mathcal{G}$ is used to combine the output distributions from the $k$ different decoders. Let $h_i^m$ be the hidden state output of the $m^{th}$ decoder layer at time step $i$, i.e. the output of the last shared layer. We use this hidden state representation to obtain the coefficients for our mixture of experts. The representation $h_i^m$ is fed into a feed forward layer $W$ (size $= (|h_i^m|, k)$), followed by a softmax layer. This outputs a probability distribution $g_i$ which is used to compute the overall next-token output probability as follows: $P(y_i|x, y_{<i}) = \sum_{j=1:k} g_i^j * P_{\phi_j}(y_i|x, y_{<i})$. Here, $g_i^j$ is the probability of selecting the $j^{th}$ decoder at time step $i$.

## 2.1 TRAINING

The HYDRASUM architecture is trained to minimize the cross entropy loss of the reference summaries, conditioned on the input document: $loss = -\sum_i \log P(y_i|x, y_{<i})$. This is same as the training objective of standard summarization models. We consider two different variants within this framework which differ in how the gates $g$ are derived (see Figure 2):

Under the **unguided training** setting, no additional supervision is provided to the gating mechanism. The model implicitly decides the contribution of each decoder to the final output probability, i.e. $g_i^j$ for decoder $j$ at time step $i$, using the gating mechanism $\mathcal{G}$ outlined above. This mixture-based formulation allows the HYDRASUM model to assign contrasting summary styles to different decoders as well as learn from infrequent examples in the dataset. Experiments under the unguided setting are outlined in Section 3.1.

Under the **guided training** setting, we explicitly guide which summary feature is partitioned between the multiple decoders (we only evaluate the 2-decoder version of HYDRASUM under this setting). Given a target stylistic feature, say specificity, our goal is to ensure that the two decoders generate summaries that vary substantially along this style. More concretely, we want decoder 0 and decoder 1 to generate summaries with low and high levels of specificity respectively. To achieve this, we first derive multiple partitions of the training data based on the specificity of the reference summary (or sentence); let $g \in [0, 1]$ denote specificity. Next, instead of using the gating mechanism $\mathcal{G}$ during training, we use this oracle label $g$ to derive the mixture coefficients $[1 - g, g]$. In this way, the oracle label $g$ is used to decide the contribution of each decoder to the final output probability and loss computation, which is modified as follows:

$$loss = -\sum_i \log\left[(1 - g) * P_{\phi_0}(y_i|x, y_{<i}) + g * P_{\phi_1}(y_i|x, y_{<i})\right]$$

Here, if $g = 0$, it dictates that only decoder 0 is used, if $g = 1$, only decoder 1 is used. By setting $g \in (0, 1)$, we can train the models using a mixture of decoders to denote mid-level specificity. More details about the training procedure and experiments under this setting are outlined in Section 3.2.

## 2.2 INFERENCE

The multi-decoder framework of HYDRASUM provides several options of output probability distributions which differ in how the mixture weights are obtained (see Figure 2). During inference, we can sample from these different options, or inference strategies, to generate diverse summaries:

**(Inference Strategy 1) Individual Decoders**: To generate summaries using only the $j^{th}$ decoder, the output of the gating mechanism is overridden with $[0, 0..., 1, ..., 0]$ where $g^j = 1$ and $g^{i \neq j} = 0$ for all time steps.

**(Inference Strategy 2) Mixture using $\mathcal{G}$**: The mixture weights are decided by the model, i.e. $g_i^j = (W^T h_i^m)_j$ for decoder $\phi_j$ at time step $i$. Note that this inference strategy can only be used for HYDRASUM models trained using the unguided setting.

**(Inference Strategy 3) Mixture using manually-specified g**: Consider a 2-decoder HYDRA-SUM model, where decoder 0 learns abstractive features and decoder 1 learns extractive features. The user can control the degree of abstraction in the generated summaries by sampling the model using different values of $[1 - g, g]$. For e.g., the user-specified distribution $[0.3, 0.7]$ gives the following output probability: $P(y_i|\cdot) = 0.3 * P_{\phi_0}(y_i|\cdot) + 0.7 * P_{\phi_1}(y_i|\cdot)$.

## 3 EXPERIMENTS

We perform experiments on three news summarization datasets: CNN (Hermann et al., 2015; Nallapati et al., 2016), NEWSROOM[2] (Grusky et al., 2018) and XSUM (Narayan et al., 2018). For all experiments, BART-LARGE (Lewis et al., 2020) is used as the model initialization: in a $k$-decoder variant of HYDRASUM, all $k$ decoders are initialized with the weights of the BART-LARGE decoder. The weights of the gating mechanism $\mathcal{G}$ are randomly initialized from a normal distribution $\mathcal{N}(0, 0.02)$. All our experiments are conducted using the 2-decoder version of HYDRASUM, setting the number of shared layers, i.e. $m$ to 8. Our model architecture is implemented using the Huggingface Library (Wolf et al., 2020). Further details about the training data and hyperparameters are included in Appendix A.

We consider the standard BART-based summarization model as our baseline model. For XSUM, we use the publicly available BART-LARGE-XSUM checkpoint. For CNN and NEWSROOM, we fine-tune the BART-LARGE checkpoint on their corresponding training datasets ourselves.[3] Beam decoding is used to generate summaries for both the baseline and proposed models.

In the following subsections we describe experiments conducted in the Unguided and Guided training settings as outlined in Section 2.1.

### 3.1 UNGUIDED TRAINING

**STYLE PARTITIONING AND EVALUATION** First, we aim to answer the following question: Do individual HYDRASUM decoders learn different summary styles when trained using the standard training objective, i.e. the unguided training setting outlined in Section 2.1? If yes, which stylistic features vary across multiple decoders?

To answer this, we measure the differences between generated summaries along the following stylistic features: 1) **Abstractiveness**: We follow Grusky et al. (2018) and report two metrics for abstractiveness, *coverage* which denotes the fraction of words in the summary that are present in the input article, and *density* which denotes the average length of text spans in a summary that are copied from the input article. Details about these metrics can be found in the original paper. Additionally, we also report the 2-gram overlap between the generated summary and the input article. 2) **Specificity**, quantified using the Speciteller tool (Li & Nenkova, 2015), to measure the degree of specificity vs generality of the summaries. To align with the specifications of the tool, we segment summaries into sentences and report the macro-average of the sentence-level specificity across all summaries. 3) **Length metrics**: We report two metrics for this, *absolute length* (number of words) of generated summaries, and *compression ratio*, computed as the ratio of the number of words in the summary

---

[2] We conduct experiments on the *mixed* subset of NEWSROOM to limit the dataset size. We found that summaries in this subset were less noisy and more diverse than the *abstractive* and *extractive* subsets respectively.

[3] We found that publicly available BART-LARGE-CNN (Lewis et al., 2020) and PEGASUS-NEWSROOM (Zhang et al., 2020) models trained on the entire CNNDM and NEWSROOM datasets performed poorly on the CNN only and NEWSROOM-MIXED only test sets used in our paper. Therefore, we re-train these.

| | Coverage | Abstractiveness | | Specificity | Length-metrics | | Readability | Quality |
| | | Density | 2G Overlap | | Abs. Len | Comp. Ratio | FRE | R1/R2/RL |
|---|---|---|---|---|---|---|---|---|
| **CNN** | | | | | | | | |
| Ref | 0.85 | 3.14 | 0.43 | 0.44 | 37.33 | 0.07 | 52.51 | - |
| Baseline | 0.97 | 10.33 | 0.80 | 0.44 | 50.71 | 0.10 | 54.03 | 34.87/14.88/31.82 |
| D0 | 0.93 | 5.69 | 0.64 | 0.48 | 46.07 | 0.09 | 58.00 | 34.58/13.64/31.43 |
| D1 | 0.97 | 11.69 | 0.82 | 0.40 | 59.47 | 0.11 | 50.92 | 31.44/11.72/28.58 |
| Mix | 0.97 | 11.1 | 0.81 | 0.46 | 54.66 | 0.10 | 53.7 | **34.91/14.36/31.93** |
| **NEWSROOM** | | | | | | | | |
| Ref | 0.83 | 3.40 | 0.46 | 0.57 | 23.67 | 0.07 | 50.8 | - |
| Baseline | 0.96 | 14.34 | 0.80 | 0.63 | 34.11 | 0.10 | 48.64 | **36.38/19.54/31.20** |
| D0 | 0.90 | 6.15 | 0.59 | 0.65 | 33.95 | 0.10 | 49.58 | 34.64/16.59/28.94 |
| D1 | 0.96 | 16.45 | 0.84 | 0.58 | 34.66 | 0.10 | 49.41 | 33.73/17.27/28.90 |
| Mix | 0.96 | 17.13 | 0.81 | 0.63 | 38.34 | 0.11 | 48.38 | 35.32/18.69/30.31 |
| **XSUM** | | | | | | | | |
| Ref | 0.66 | 1.05 | 0.16 | 0.65 | 21.1 | 0.09 | 59.6 | - |
| Baseline | 0.75 | 1.61 | 0.27 | 0.56 | 19.20 | 0.09 | 66.70 | **45.14/22.27/37.25** |
| D0 | 0.72 | 1.37 | 0.23 | 0.66 | 19.72 | 0.09 | 60.45 | 42.82/19.16/34.15 |
| D1 | 0.72 | 1.44 | 0.23 | 0.53 | 19.96 | 0.09 | 62.70 | 42.33/18.56/33.98 |
| Mix | 0.73 | 1.51 | 0.25 | 0.59 | 19.60 | 0.10 | 62.07 | 44.72/21.47/36.36 |

Table 1: Comparison of HYDRASUM's generated summaries using individual decoders (D0 and D1) and their model-derived mixture (Mix). Results show significant differences along multiple dimensions (highlighted in gray), most notably abstractiveness and specificity for CNN and NEWSROOM, and specificity for XSUM.

and the input article. 4) **Readability**: Finally, we report the readability scores of the summaries, measured using the Flesch readability ease test (Flesch, 1948). In addition to these style-based metrics, we report **Quality**, measured by ROUGE (Lin, 2004) scores of the generated summaries with respect to the reference summaries. For analysis, we generate 3 summaries for each input: using individual decoders D0 and D1 (Inference Strategy 1, see Section 2.2), and the mixture model (Mix) where the mixture weights are obtained using the gating mechanism $\mathcal{G}$ (Inference Strategy 2). The latter strategy corresponds to sampling from the HYDRASUM model's actual output distribution. Results of the study are shown in Table 1.

**Style differences between decoders**: We study the difference in stylistic features between D0 and D1. Features for which this difference is significant, i.e. $p < 0.05$ according to the bootstrap re-sampling test, are highlighted in gray. For both CNN and NEWSROOM, significant differences are observed along the abstractiveness and specificity metrics. Moreover, summaries for CNN also differ along other metrics such as length and readability. Interestingly, for both these datasets, we observed that the baseline model fails to cover the entire range of abstractive behavior seen in the reference summaries. Figure 4 demonstrates this; the top graphs plot the 2-gram overlap of the reference summaries and the generated summaries for the baseline model, showing substantial mismatch. This phenomenon has been discussed in prior work (See et al., 2017); summarization models tend to overfit on the easier extractive examples, and do not learn from the abstractive examples. HYDRASUM addresses this limitation by encouraging the two decoders to learn contrasting levels of abstractiveness. Figure 4 shows D0 decoders for both datasets learn to generate abstractive summaries that more closely resembles the reference distribution. Meanwhile, D1 generates abstractive summaries, collectively providing better coverage over the abstractiveness space.

The results also indicate that the least amount of style difference is observed for the XSUM-based HYDRASUM model. Here, significant variance is observed only along specificity: D0 generates more specific and D1 generates more general summaries. However, the observed difference in specificity for XSUM (approx. .13) is greater than that of the other two datasets. We hypothesize that the similarity in abstractiveness levels between D0 and D1 is due to the low diversity along this feature in XSUM's reference summaries. The results in Table 1 indicate that although the HYDRASUM model's training encourages the two decoders to learn distinct stylistic features, the combination of features along which they differ is heavily dependant on the datasets themselves. In Section 3.2, we evaluate a more controlled version of our model by explicitly guiding which single style feature differs between D0 and D1 and encouraging higher diversity along that target feature.

**Quality**: The ROUGE scores of the generated summaries using the entire HYDRASUM model, i.e. Mix, are comparable to the baseline BART models, even outperforming the baseline for CNN. This

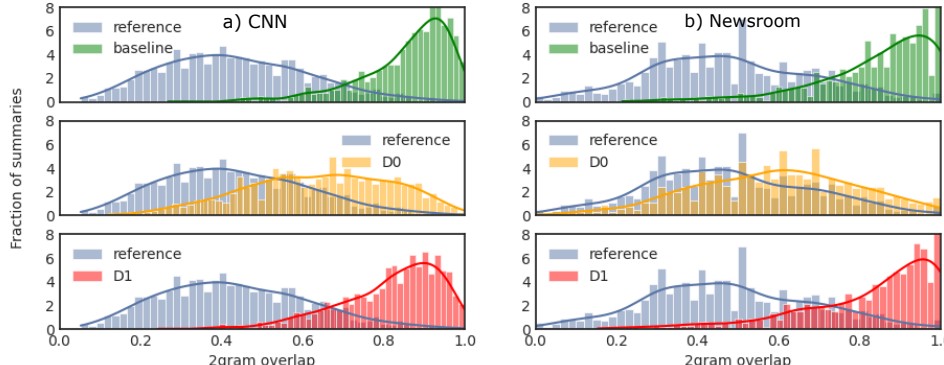

Figure 4: Graphs plot the 2gram overlap of the baseline and HYDRASUM decoders. Compared to the baseline, D0 decoder samples summaries from a distribution that more closely resembles the reference distribution.

| Model | CNN | | | NEWSROOM | | | XSUM | | |
|---|---|---|---|---|---|---|---|---|---|
| | TopK-ROUGE | $\sigma$(ov.) | $\sigma$(sp.) | TopK-ROUGE | $\sigma$(ov.) | $\sigma$(sp.) | TopK-ROUGE | $\sigma$(ov.) | $\sigma$(sp.) |
| Base + Beam | 39.10/17.76/35.65 | .03 | .07 | 43.00/24.73/36.98 | .06 | .10 | 50.19/**25.74**/40.86 | .04 | .07 |
| + Top-k | 40.29/15.37/36.14 | **.12** | **.13** | 43.58/22.25/36.27 | **.16** | **.21** | 48.16/21.68/37.98 | **.07** | **.16** |
| + DBS | 40.62/18.65/37.04 | .05 | .08 | 43.59/24.72/37.27 | .07 | .12 | 50.52/25.72/41.06 | .04 | .08 |
| HS + Beam | **42.07/19.19/38.32** | .11 | .11 | **45.03/25.59/38.46** | .14 | .15 | **51.03**/25.46/**41.18** | .06 | .13 |

Table 2: Diversity performance of the baseline BART models (Base) and HYDRASUM (HS).

demonstrates that the inclusion of additional decoders in HYDRASUM's architecture does not hurt the quality of generated summaries. On the other hand, the quality of summaries generated from individual decoders is roughly 2 ROUGE points lower than both the Mix strategy and the baseline model. This is expected; individual decoders generate summaries that exhibit "extreme" or contrasting behaviors along different style features (shown above). Therefore, they under perform when evaluated on the entire test set containing a diverse set of styles.

**DIVERSITY EVALUATION** Next, we quantitatively evaluate whether the mixture-of-experts formulation of HYDRASUM leads to higher diversity in multiple ($> 2$) summary generation scenarios compared to the baseline models.

We report 2 metrics to evaluate diversity: 1) **TopK ROUGE**: The maximum ROUGE (R1/R2/RL) score over a list of K generated summaries for a given input document. This gives an upper bound on the benefit from diverse summarization by measuring the closeness of the best generated summary to the reference summary. 2) **Stylistic Diversity**: The standard deviation within the K generated summaries of a given example, along each style metric independently. This metric indicates the average variety in summary choices available to the user for each input. Results are reported for abstractiveness (2 gram overlap, ov.) and specificity (sp.). We set K= 5 for our experiments. For HYDRASUM, multiple summaries are generated by varying the summary-level gating probability $g$ as outlined in Section 2.2: Strategy 3. We use $g = \{0, 0.25, 0.5, 0.75, 1\}$. Here $g = 0$ and $g = 1$ correspond to summaries generated using decoder 1 and decoder 0 independently. We compare these to K summaries of the baseline BART model generated using three different decoding strategies: beam search, top-k sampling, and diverse beam search (Vijayakumar et al., 2018). Decoding hyperparameters for both baseline and HYDRASUM models are included in Appendix A.

Table 2 outlines the diversity performance of the baseline model and HYDRASUM. The results show that HYDRASUM substantially outperforms the baseline's TopK-ROUGE performance, across different decoding strategies considered. In fact, the gain is roughly proportional to the degree of stylistic difference observed in Table 1; the highest gain (roughly +3 ROUGE points) is reported for CNN, followed by an improvement of +2 ROUGE points for the NEWSROOM dataset. In terms of stylistic diversity, our results show that the HYDRASUM model outperforms the baseline model when decoded using the same decoding strategy, i.e. beam search, for all three datasets. On the other hand, we see that top-k decoding leads to more style diversity within baseline summaries. However, this is not accompanied by a corresponding increase in quality, which indicates that this additional diversity is achieved by sampling low quality summaries.

**QUALITATIVE ANALYSIS** Finally, we qualitatively evaluate the style difference between summaries generated using individual decoders. Figure 5 provides examples of generated summaries using HYDRASUM trained on the NEWSROOM dataset. For the first example, consistent with the

| Input Article | D0 Summary | D1 Summary |
|---|---|---|
| Forget gold and oil. Copper prices is the real winner this year. The red metal is up more than 20 percent from its late January low — and that's given one stock a big boost: Freeport-McMoRan. The mining giant is up 40 percent in the same period, but one trader who relies heavily on the technicals and options market, is cautious on the stock, and he warned that the rally could be over. ``I think we're about to see some serious selling pressure in Freeport'' said technical analyst Andrew Keene on CNBC's `` Trading Nation'' on Thursday. [...] | Copper prices are up 20 percent this year, and one miner is up 40 percent. But one trader warns that the rally could be over. | Copper prices is the real winner this year. The red metal is up more than 20 percent from its late January low — and that 's given one stock a big boost. |
| Jenson Button and his wife Jessica have been robbed at a holiday home in Saint-Tropez. (AAP) - British Formula One star Jenson Button and his model wife Jessica Michibata are believed to have been knocked out with gas during a brazen robbery in which thieves made off with more than A$ 630,000 worth of their possessions. The couple were in a rented villa in the glitzy French coastal resort of Saint-Tropez with friends when the bandits struck. [...] | **British Formula One driver** Jenson Button and his wife **Jessica Michibata** have been robbed at their holiday home in Saint-Tropez. | Jenson Button and his model wife have been robbed at their holiday home in Saint-Tropez . |

Figure 5: Examples of generated summaries using the unguided setting for NEWSROOM dataset. Long extractive sequences are underlined, additional details that increase the specificity of summaries are in bold.

observation from Table 1, D1 generates a highly extractive summary whereas D0 generates an abstractive summary with less copying. In the second example, we observe a difference in specificity of the generated summaries. D0 summary includes additional details like *Jenson Button*'s profession and his wife's name, compared to the more less specific summary generated by D0.

Note that additional experiments with analysis of the 3-decoder HYDRASUM model is included in Appendix B. Also, experiments with other values of $m(= 6, 10)$ are in Appendix C. We found that the choice of $m$ does not significantly alter our analysis.

### 3.2 GUIDED TRAINING

Here, we train HYDRASUM models such that given a target style, decoders D0 and D1 learnt contrasting or "extreme" behaviors for that style, e.g. very extractive vs very abstractive summaries. We consider two features for our experiments in this section: abstractiveness (measured by 2-gram overlap) and specificity. Let $f$ denote the target feature. To ensure D0 learns low-$f$ and D1 learns high-$f$, we carefully control the subset of the training data used to train each decoder. Our exact methodology is as follows: 1) First, we pre-process the training data to derive $n(= 5)$ percentile splits based on the $f$-value of reference summaries. For e.g., if $f$ refers to abstractiveness, we split the training data based on 2-gram overlap. 2) Next, we modify the loss computation for each example during training to incorporate information about the percentile split it belongs (refer to Section 2.1 for the modified loss function). In our experiments, we set $g \in \{0, 0.25, 0.5, 0.75, 1\}$. Effectively, this controls the contribution of each decoder in a training example's final loss. For e.g., the bottom 20 percentile split of the data (low $f$) are trained by setting $g = 0$, i.e. using only D0. This ensures that D0 learns to generate low-$f$ summaries. Note that the oracle gate $g$ can be defined at the token-, sentence- or summary-level in the above equation. Since the specificity metric is defined at the sentence-level, we use oracle gates $g_t$ that denotes the gate for sentence $s_t$.

**SINGLE-FEATURE CONTROL** First, we evaluate if HYDRASUM can enforce higher stylistic variation between the individual decoders for a given target feature compared to the unguided setting.

To answer this, we train and evaluate models for 2 target features: abstractiveness ($f$ = abstractiveness) and specificity ($f$ = specificity), where $f$ refers to the target feature. For each model, we report the following metrics: (1) $f$(D0) and $f$(D1): The average style scores for test summaries generated using D0 and D1 respectively. This refers to the 2-gram overlap when $f$ = abstractiveness and specificity when $f$ = specificity. (2) TopK $\sigma(f)$: Similar to Section 3.1, we generate 5 summaries by varying gate probabilities: $g = \{0, 0.25, 0.5, 0.75, 1\}$, and report the standard deviation of the style score $f$ amongst these 5 summaries.[4] Results are outlined in Table 3.

The results show that compared to the the unguided setting (see Table 1), the guided training approach enforces a substantially higher difference in target style $f$ between D0 and D1. For e.g., consider the specificity-controlled model for CNN. Under the guided setting, the specificity difference between D0 and D1 is roughly .40 points, compared to .08 for the unguided models. Similar improvements are observed across all models and style combinations. Guided training even succeed in enforcing abstractive style variance for XSUM models; this was not possible under the unguided setting. Finally, the results show significant improvement in the stylistic diversity within the top

---

[4]The entire set of results, including ROUGE, TopK-ROUGE, and other style scores are in Appendix D.

| Metric | $f$ =Abstractiveness | | | $f$ =Specificity | | |
|---|---|---|---|---|---|---|
| | CNN | NEWSROOM | XSUM | CNN | NEWSROOM | XSUM |
| $f(D0)$ / $f(D1)$ | .48 / .82 | .44 / .85 | .16 / .29 | .22 / .62 | .36 / .81 | .44 / .80 |
| TopK $\sigma(f)$ | .14 | .17 | .07 | .16 | .21 | .16 |

Table 3: Performance of HYDRASUM models in the guided setting. Compared to the unguided setting, we observe higher variation in style between D0 and D1 as well as better TopK style diversity.

| **Low Specificity Decoder (D0)** | **High Specificity Decoder (D1)** |
|---|---|
| Two Florida boys are being hailed as local heroes after saving children from a burning mobile home | Isiah Francis, 10, and Jeremiah Grimes, 11, saved two babies from a burning mobile home in Florida. |
| French prosecutor says he is not aware of any video footage from on board the plane. | French prosecutor says he's not aware of any video footage from on board Germanwings Flight 9525. |

Table 4: Example summaries generated using low and high specificity decoders in the specificity-guided setting. The underlined text highlights additional details in the more specific summaries.

5 summaries over the unguided setting (compare with results from Table 2). Most notably, the specificity-controlled NEWSROOM model exhibits $\sigma(sp.)$ of .21 within generated summaries compared to .15 in the unguided setting. Table 4 gives examples of high and low-specificity summaries.

Next, we study the stylistic properties of the 5 summaries generated by varying $g$, used in the above experiments. Ideally, we want these summaries to exhibit stylistic behavior (value of average style score $f$) between that of D0 and D1. Concretely, since D0 generates low-$f$ summaries and D1 generates high-$f$ summaries, we want summaries generated by setting $g = 0.5$ to result in mid-level $f$ scores. To study this, we plot the 2gram overlap of CNN summaries for the 5 gate values ($g = \{0, 0.25, 0.5, 0.75, 1\}$) for the abstractiveness-controlled model (top row of Figure 6). Similarly, we plot specificity distributions for specificity-controlled model corresponding to these different gate probabilities (bottom row of Figure 6). Due to space constraints, graphs for the NEWSROOM and XSUM are in Appendix D. For both these stylistic features, we observe that the HYDRASUM model shows a gradual increase in average feature scores as the contribution of D1 (high-$f$ decoder) is increased, from 0 contribution in the left-most graphs to 1 in the rightmost graphs. This shows that HYDRASUM can be used by end users to generate summaries corresponding to their desired degree of abstractiveness or specificity, by selecting and appropriate value of the gate probability $g$.

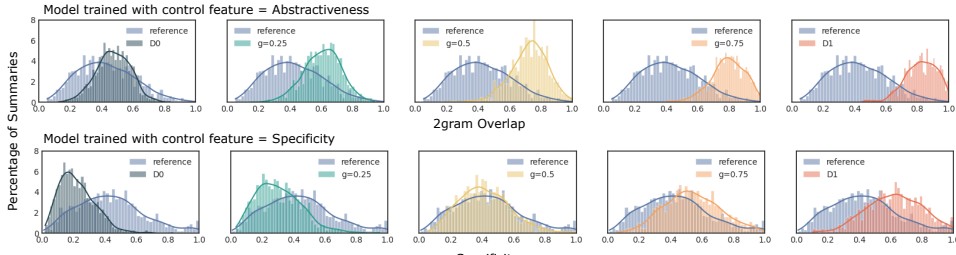

Figure 6: 2gram overlap and specificity of CNN summaries generated using different values of $g$ in the guided setting. The top graphs are obtained by mixtures of decoders from the abstractiveness-controlled model; the bottom graphs are from the specificity-controlled model. These graphs provide evidence that properties like abstractiveness and specificity can be varied by varying the gate probabilities in the guided setting.

**MULTI-FEATURE CONTROL** Next, we investigate the multi-feature control capabilities of HYDRASUM. We study whether the decoders from separate single-feature controlled models, corresponding to orthogonal features such as abstractiveness and specificity, be combined to exhibit multi-feature control over generation.

Due to the low abstractive variance in XSUM, we only perform these experiments on CNN and NEWSROOM datasets. For these, we use a combination of the abstractive/extractive and specific/general decoders to obtain 4 different summaries, setting gate probability $g = 0.5$. We plot the marginal distributions for each feature $f$ for both datasets in Figure 9. The graphs show that the HYDRASUM model ensures that summaries generated using the high specificity decoder have higher average specificity than those that include contribution from the low specificity decoders. Similar trends are observed with respect to abstractiveness. Note that the models used in this experiment, i.e. the abstractiveness and specificity controlled models for the two datasets were trained independently.

Figure 7: 2gram overlap and specificity of CNN and NEWSROOM summaries generated using a combination of specificity-controlled and abstractiveness-controlled decoders.

| | CNN | | NEWSROOM | | XSUM | |
|---|---|---|---|---|---|---|
| | $f = $ Abs. | $f = $ Spec. | $f = $ Abs. | $f = $ Spec. | $f = $ Abs. | $f = $ Spec. |
| HYDRASUM D0 | **4.4/4.5/4.3**/.93 | **4.4**/4.3/4.2/.85 | **4.3/4.4/4.1**/.9 | 4.1/4.2/3.5/.80 | 4.2/4.3/4.1/.89 | 4.3/4.4/4.1/.81 |
| HYDRASUM D1 | 4.3/**4.5/4.3**/.89 | **4.4**/4.3/4.1/.87 | 4.2/4.2/4.0/.9 | 4.2/**4.4/4.1**/.85 | 4.3/**4.5**/4.0/.87 | **4.4**/4.4/**4.2**/.89 |
| Baseline BART | 4.3/4.4/4.2/.83 | | 4.2/4.3/4.0/.85 | | 4.3/4.4/**4.2**/0.85 | |

Table 5: Comparison of human-rated **Relevance/Coherence/Grammaticality/Factuality** scores for Abstractiveness- and Specificity-controlled HYDRASUM models and the baseline BART model.

This demonstrates the high flexibility of HYDRASUM's architecture: it allows mixing decoders from independently trained models at inference time and generates summaries corresponding to multiple control features.

**HUMAN EVALUATION** Following prior work (Hashimoto et al., 2019), we rely on automated metrics for diversity evaluation and conduct human evaluation to measure the quality of generated summaries. For 50 randomly sampled input articles, we present MTurk workers with 5 different generated summaries: baseline model summary, D0 and D1 summaries of the Abstractiveness-Specificity-controlled models. The workers were asked to rate each summary across 4 properties: relevance, coherence, grammatically and factuality. For the first 3, we ask for a rating on the 5-point Likert scale. Following prior work (Goyal & Durrett, 2021), we seek binary labels (factual (1) or non-factual(0)) for factuality annotation. We report the average score across all three annotations. Table 11 outlines the results. Across all metrics, we see that the humans prefer summaries generated by the HYDRASUM models more than the baseline models. More details about the human evaluation task setup as well as results for the unguided setting are in Appendix F.

## 4 RELATED WORK

Previous work on controllable summarization has focused on single-feature control of length (Fan et al., 2018a; Saito et al., 2020; Makino et al., 2019), entities or topics (He et al., 2020), or abstractiveness (Song et al., 2020). However, these methods are over-specialized for their target feature, and cannot be generalized to other control dimensions or multi-feature control. In our paper, we focus on *style* control instead of *content* control, and propose a generalizable approach that can be adapted to any style feature. Recently, Song et al. (2021) proposed a new technique for control that involves over-generation and post-filtering. Our model architecture can automatically identify diverse styles within the training data, and disentangles them to generate a diverse set of summaries. Finally, style controlled generation has also been studied for other generation tasks such as paraphrasing, story generation and machine translation (Wang et al., 2017; Shen et al., 2017; Huang et al., 2019; Yang & Klein, 2021). These require significant changes to the baseline architecture; we show that style control can be achieved using minimal changes to either the architecture or the loss function.
Diverse generation research has focused on lexical diversity (Vijayakumar et al., 2018; Kumar et al., 2019), syntactic diversity (Goyal & Durrett, 2020), or through uninterpretable latent codes (Park et al., 2019; Shao et al., 2019). In this work, we study a more controlled version of diversity enforcement along specific target style features.

## 5 CONCLUSION

We propose a new summarization architecture HYDRASUM containing multiple decoders in a mixture-of-experts framework. Through experiments on 3 summarization datasets, we show that the proposed models can effectively disentangle distinct stylistic features, such as high or low abstractiveness, different degrees of specificity, etc. during summary generation. Moreover, our framework is highly flexible: during inference, we can sample from either individual decoders or their mixtures to generate diverse summaries and enforce both single- and multi-style control over generation.

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

## A    TRAINING DETAILS

We evaluate our models on three datasets: CNN, NEWSROOM and XSUM. Details about the training, development and test dataset sizes for these are outlined in Table 6. Note that our experiments (both training and testing) are performed on the *mixed* subset of the NEWSROOM dataset. All results and analysis in the paper is reported on the test data.

| Dataset | Training | Dev | Test |
|---------|----------|-----|------|
| CNN | 90266 | 1220 | 1093 |
| NEWSROOM | 329494 | 35977 | 36100 |
| XSUM | 204045 | 11332 | 11334 |

Table 6: Dataset statistics

Table 7 outlines the hyperparameters used for training and inference. For all our experiments, we use BART-LARGE as the pre-trained initialization. During inference for HYDRASUM, we incorporate top-k and top-p sampling using values 30 and 0.5 respectively. For top-k decoding for using baseline BART model in Table 2, we set $k = 30$. Diverse beam search is run using 2 beam groups and diversity penalty 0.5.

## B    EFFECT OF DIFFERENT NUMBER OF DECODERS

Next, we investigate the effect of the number of decoders $k$ on the partitioning of summary styles by extending our analysis to a 3-decoder variant of HYDRASUM. Table 8 outlines our results. For simpler analysis, we only report results along 4 metrics: ROUGE score, 2-gram overlap, specificity and absolute length. Similar to the 2-decoder case, the 3 decoders of HYDRASUM learn a mutually-distinct combination of of summary styles. In fact, we observe that 3-way partitioning enables the model to cover a wider variety of summary styles. For example, two XSUM decoders learn to generate relatively more extractive and longer summaries (D0 and D2). While enforcing extractiveness is not straightforward for baseline BART models or the XSUM decoders in Table 1, we can now sample from D0 or D2 to obtain more extractive summaries. Similarly, the range of specificity provided by CNN ($.34 - .55$) and NEWSROOM ($.42 - .67$) models is higher than that of the 2-decoder variant. Finally, in Figure 4, the graphs show that for both CNN and NEWSROOM, the baseline models fail to cover the entire spectrum of abstractiveness exhibited in the training set due to overfitting on easier extractive examples. The results in Table 8 shows that using multiple-decoders allows HYDRASUM models to learn from such minority examples and exhibit the entire breadth of behaviors seen

| | For training | | For Inference | |
|---|---|---|---|---|
| Implementation | Huggingface (Wolf et al., 2020) | | CNN & NEWSROOM | |
| Computing Infrastructure | 40 GB NVIDIA A100 GPU | | Num beams | 5 |
| Optimizer | Adam | | Length Penalty | 2 |
| Optimizer Params | $\beta = (0.9, 0.999), \epsilon = 10^{-8}$ | | No repetition size | 3-grams |
| Learning Rate Decay | Linear | | Min-Length | 12 |
| Learning rate | 1e-5** | | Max Length | 200 |
| Weight Decay | 0 | | XSUM | |
| Maximum Gradient Norm | 1 | | Num beams | 6 |
| Batch size | 64 | | Length Penalty | 1 |
| Epochs | 3 | | No repetition size | 3-grams |
| Max Input Length | 1024 (512 for NEWSROOM) | | Min-Length | 12 |
| Max Output Length | 128 | | Max Length | 60 |

Table 7: Hyperparameters used from fine-tuning and decoding the BART-based summarization models. (**For specificity-controlled models in Section 3.2, we employ a learning rate of 2e-5)

| Dataset | Decoder | Rouge (R1/R2/RL) | Overlap | Spec. | Length | Characteristics |
|---|---|---|---|---|---|---|
| CNN | D0 | 32.35/10.90/29.29 | .48 | .34 | 39.9 | Low Copy, Low Spec. |
| | D1 | 21.63/8.48/20.18 | .82 | .38 | 180.7 | High Copy, Low Spec. |
| | D2 | 33.86/13.23/30.87 | .72 | .55 | 56.1 | Avg. Copy, High Spec. |
| | Mix | **34.30/14.38/31.36** | .82 | .48 | 56.2 | High Copy, Avg Spec. |
| NEWSROOM | D0 | 31.88/14.71/27.12 | .32 | .42 | 32.0 | Low Copy, Low Spec. |
| | D1 | 16.05/6.94/14.39 | .36 | .49 | 171.9 | Avg. Copy, Avg. Spec. |
| | D2 | 32.43/16.57/27.61 | .85 | .67 | 47.9 | High Copy, High Spec. |
| | Mix | **35.39/18.85/30.37** | .82 | .64 | 38.9 | High Copy, High Spec. |
| XSUM | D0 | 31.63/12.21/24.83 | .36 | .60 | 44.6 | High Copy, Avg. Spec. |
| | D1 | 41.86/17.97/33.22 | .22 | .54 | 20.1 | Low Copy, Low Spec. |
| | D2 | 32.33/12.63/25.44 | .32 | .67 | 44.1 | High Copy, High Spec. |
| | Mix | **44.61/20.91/36.17** | .24 | .58 | 19.5 | Low Copy, Avg. Spec. |

Table 8: Comparison of generated summaries for a 3-decoder HYDRASUM model. Results show higher coverage of summary styles by individual decoders compared to the 2-decoder version. For e.g., D0 and D2 of the XSUM model learn to generate relatively more extractive and longer summaries; partitioning along abstractiveness and length was not observed in the 2-decoder version (see Table 1).

in the training data. Finally, we see that some decoders report very poor quality (ROUGE scores). This is expected due to two factors: 1) since these decoders learn from minority examples, they exhibit less common summary styles and suffer on dataset-wide evaluation, and 2) factors such as very longer length directly affects the precision of ROUGE scores. However, note that across all datasets, mixture-based inference performs best. This shows that although the performance of the individual decoders is low, their contribution to the mixture is needed for the overall best performance.

## C EFFECT OF DIFFERENT NUMBER OF SHARED LAYERS

In order to restrict the number of extra parameters introduced in HYDRASUM, we enforced parameter sharing between the $m$ lower layers of the decoders. We performed our experiments in Section 3 by setting $m = 8$. Here, we investigate if the choice of $m$ effects either the partitioning of stylistic features between decoders, or the extent of the observed difference between two decoders along any axis such as abstractiveness, specificity, etc. Experiments are performed using the 2-decoder version of HYDRASUM. We train models using the unguided setting (same as Section 3.1) for $m = 6, 10$ for all 3 datasets. For simpler analysis, we only report on a subset of the metrics: ROUGE scores (quality), 2 gram overlap (abstractiveness), specificity, absolute length, and self-ROUGE between the summaries generated using individual decoders (diversity).

| m | ROUGE | | Overlap | | Specificity | | Length | |
|---|---|---|---|---|---|---|---|---|
| | D0 | D1 | D0 | D1 | D0 | D1 | D0 | D1 |
| CNN | | | | | | | | |
| 6 | 33.21/13.3/30.21 | 34.26/13.3/31.21 | .79 | .63 | .35 | .41 | 44.9 | 54.5 |
| 10 | 32.04/12.37/29.13 | 35.2/14.11/32.19 | .80 | .68 | .36 | .44 | 53.8 | 45.9 |
| NEWSROOM | | | | | | | | |
| 6 | 32.32/16.17/27.5 | 34.92/17.05/29.55 | .82 | .61 | .53 | .60 | 39.5 | 30.0 |
| 10 | 33.14/16.56/28.16 | 34.73/17.1/29.37 | .79 | .64 | .52 | .57 | 33.9 | 34.6 |
| XSUM | | | | | | | | |
| 6 | 42.2/18.7/33.6 | 42.3/18.7/33.9 | .22 | .23 | .56 | .43 | 20.2 | 19.8 |
| 10 | 42.56/19.14/34.1 | 42.83/19.15/34.24 | 23.8 | 23.1 | 54.5 | 45.8 | 19.0 | 20.5 |

Table 9: Effect of varying the number of shared layers between the 2 decoders of HYDRASUM. Results show that the choice of $m$ does not substantially alter our analysis.

Table 9 outlined the results. Compared to the HYDRASUM model variants with $m = 8$, we notice small differences in style partitioning as well as the absolute difference in style scores between decoders D0 and D1. Most notable, the CNN model with 6 shared parameters does not learn to partition across the specificity metric whereas the NEWSROOM model with $m = 6$ does learn to partition along length. These observations are different that those seen for $m = 8, 10$. However, in general, we observe that across all datasets, HYDRASUM decoders behave quite similarly in terms of which features are partitioned, irrespective of the number of shared layers $m$. This demonstrates that the proposed model architecture is useful for generating diverse summary options, even in cases where a smaller number of extra parameters are allowed.

## D GUIDED SETTING

In Section 3.2, we evaluated the diverse generation performance of HYDRASUM models under the guided setting. Table 3 outlined a brief summary of results for models trained on the three datasets. Here, we provide the entire set of results, see Table 10. In addition to the metrics reported in the main paper, we include ROUGE scores of individual decoders D0 and D1 for all $f$-controlled models. Moreover, multiple style metrics are included for each model and dataset pair (2-gram overlap, specificity and length), as well as the TopK-ROUGE scores for 5 summaries generated using Inference Strategy 3 (refer to experiment design in 3.2).

Table 10 outlines the results. In general, we observe that HYDRASUM models are able to enforce diverse generation along the target feature $f$, while limiting the stylistic variance along other features between D0 and D1. Moreover, we see that apart from XSUM, all other models report a TopK-ROUGE improvement over the baseline model performance (compare with results from Table 2).

Finally, in Figure 8, we include graphs that show the distributions of 2 gram overlap and specificity for the abstractiveness- (top row) and specificity-controlled (bottom row) models respectively. This figure includes plots for NEWSROOM and XSUM models. The corresponding graphs for CNN are included in the main body of the paper (section 3.2).

## E MULTI-FEATURE CONTROL

Figure 8 shows an example of multi-attribute control exhibited by HYDRASUM on the NEWSROOM dataset. We 4 generate summaries by using a distinct combination of extractive/abstractive and general/specific decoders from different single-feature controlled models. The figure shows the input article and these generated summaries: we see that these summary follow the style specifications of the two decoders used to construct them. Interestingly, for the High Copy, Low specificity summary, we see that the model replaces *Lyft* with *ride-sharing company* and *VanderSaden* with *former executive* from an exact copied sentence from the input, to both follow high copy and low specificity targets as faithfully as possible. In general, we found summary generation including a low specificity

| Control feature $f$ | Dec. | Quality | Summary Styles | | | Diversity-metrics | | |
|---|---|---|---|---|---|---|---|---|
| | | ROUGE | Ov. | Sp. | Len | Top5 ROUGE | $\sigma$(ov.) | $\sigma$(sp.) |
| **CNN** | | | | | | | | |
| Abstractiveness | D0 | 35.00/12.93/31.84 | .48$^\dagger$ | .42 | 48.8 | 42.50/19.10/38.69 | .14$^\dagger$ | .10 |
| | D1 | 34.66/14.45/31.78 | .82$^\dagger$ | .42 | 46.2 | | | |
| Specificity | D0 | 33.64/12.74/30.70 | .72 | .22$^\dagger$ | 48.9 | 42.05/18.84/38.23 | .09 | .16$^\dagger$ |
| | D1 | 34.40/13.35/31.18 | .69 | .62$^\dagger$ | 49.7 | | | |
| **NEWSROOM** | | | | | | | | |
| Abstractiveness | D0 | 32.56/13.98/26.68 | .44$^\dagger$ | .65 | 35.8 | 44.30/24.56/37.60 | .17$^\dagger$ | .16 |
| | D1 | 35.04/18.53/30.17 | .85$^\dagger$ | .59 | 33.9 | | | |
| Specificity | D0 | 31.62/14.80/27.11 | .67 | .36$^\dagger$ | 27.0 | 43.49/24.19/37.09 | .11 | .21$^\dagger$ |
| | D1 | 34.20/17.26/28.74 | .73 | .81$^\dagger$ | 38.4 | | | |
| **XSUM** | | | | | | | | |
| Abstractiveness | D0 | 42.45/19.00/34.35 | .16$^\dagger$ | .58 | 19.2 | 50.11/24.78/40.49 | .07$^\dagger$ | .11 |
| | D1 | 43.52/19.79/35.05 | .29$^\dagger$ | .57 | 19.5 | | | |
| Specificity | D0 | 41.84/18.55/33.86 | .22 | .44$^\dagger$ | 18.2 | 49.72/24.35/39.96 | .06 | .16$^\dagger$ |
| | D1 | 41.72/18.14/33.11 | .22 | .80$^\dagger$ | 21.8 | | | |

Table 10: Performance of style-controlled HYDRASUM models. Compared to the unguided setting, we observe higher variation in style between D0 and D1 along the control dimension (indicated with $\dagger$). Similarly, higher style diversity is observed among top 5 summaries along the control dimension.

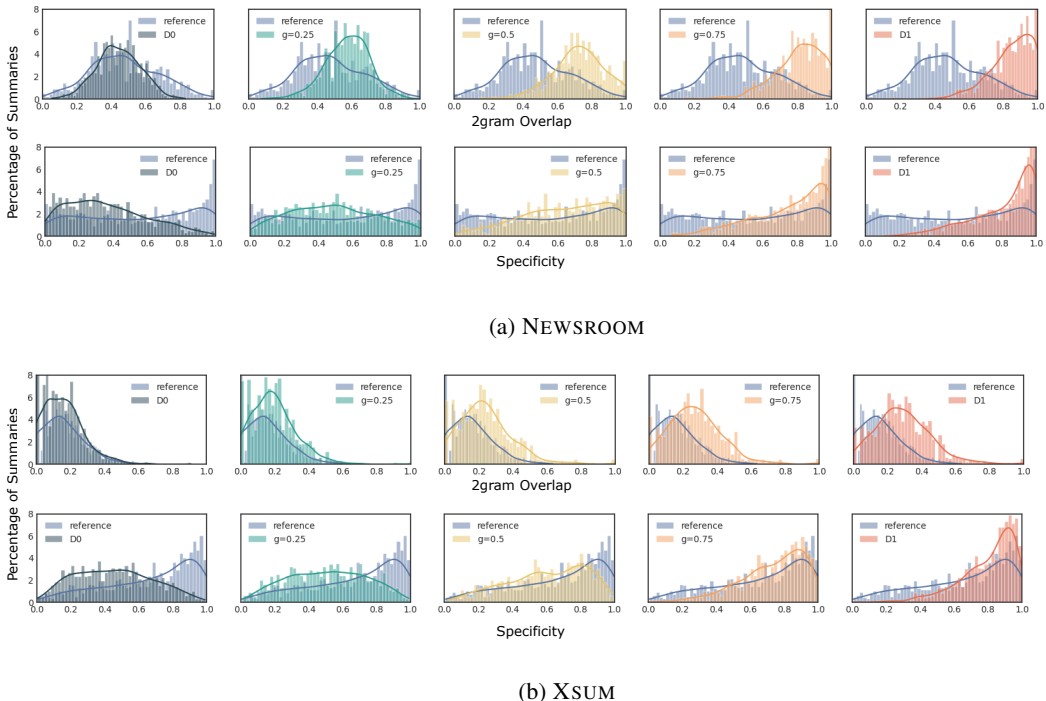

Figure 8: 2gram overlap and specificity of NEWSROOM and XSUM summaries generated using different values of $g$ in the guided setting. The graphs show that properties like abstractiveness and specificity can be controlled by sampling from a mixture of the 2 decoders corresponding to the chosen style.

decoder tougher to control (here, the Low copy, Low Specificity summary follows similar strategy to the High Copy, Low Specificity summary). This is also evidenced by specificity distributions in Figures 8 which show much higher variation for D0 (i.e. low specificity decoder) for the specificity controlled model. Similar trends are seen in Figure 9.

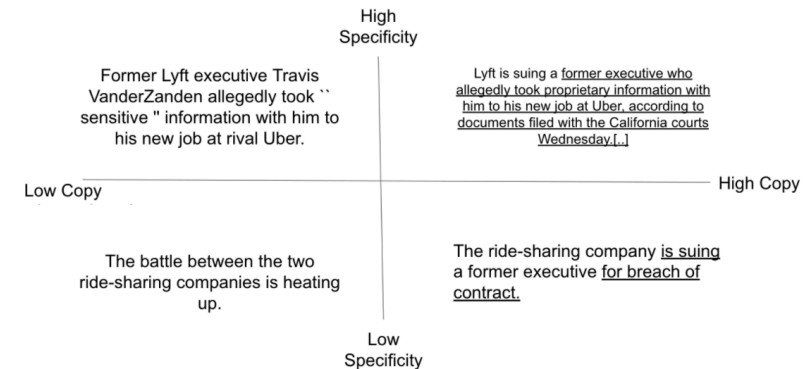

**Input Article:** The battle between Lyft and Uber is heating up -- and this time they've taken it off the road and into the courtroom. Lyft, which has been trying to expand oversees, brought a lawsuit against a former executive who allegedly took proprietary information on Lyft's international plans with him to his new job at Uber , according to documents filed with the California courts Wednesday. Travis VanderZanden previously served as chief operating officer at Lyft and left the ride-sharing company in August. He joined Uber last month as the vice president of international growth. **Lyft is suing VanderZaden for breach of contract** and said he carried `` Lyft 's most sensitive documents '' with him , which allegedly includes financial information , strategic planning , customer lists and international growth plans. [...]

Figure 9: Example of multi-feature control by HYDRASUM

|  | Baseline BART | HYDRASUM D0 | HYDRASUM D1 | HYDRASUM Mix |
|---|---|---|---|---|
| CNN | 4.4/4.4/4.2/.88 | 4.3/4.4/4.2/.86 | 4.3/4.3/4.0/.89 | 4.4/4.3/4.2/.87 |
| NEWSROOM | 4.3/4.4/4.2/.9 | 4.4/4.4/4.2/.92 | 4.2/4.4/4.1/.91 | 4.4/4.5/4.3/.90 |
| XSUM | 4.3/4.4/4.2/.77 | 4.3/4.3/4.2/.81 | 4.1/4.4/4.2/.81 | 4.2/4.5/4.3/.80 |

Table 11: Comparison of human-rated **Relevance/Coherence/Grammaticality/Factuality** scores of HYDRASUM models under the unguided setting and the baseline BART model.

## F    HUMAN EVALUATION

In section 3.2, we reported human evaluation study results under the guided setting. Here, we expand on the details of the Mechanical Turk task. Figure 10 shows task interface. For each source article, we asked 3 workers to evaluate 5 different model-generated summaries. For the unguided setting, these 5 summaries were obtained from (1) Baseline model, (2, 3) D0 and D1 decoders of the abstractiveness-controlled model, and (4,5) D0 and D1 of the specificity-controlled model. For each article-summary pair, workers were asked to rate the summaries across 4 metrics: relevance, coherence, grammaticality, and factuality. We follow prior work (Karpinska et al., 2021) and seek annotation for the first 3 on a 5-point Likert scale, with 5 corresponding to highest quality. For factuality, we ask for a binary annotation: 1 for factuality and 0 for non-factual summaries. We report the average scores of the 3 annotators across all 50 articles.

Next, we conducted an analogous study for the unguided setting. For this, we asked workers to rate the quality of 4 different summaries per article (1) baseline model, (2, 3) D0 and D1 of HYDRA-SUM model, and (4) Mix strategy of HYDRASUM model. Again, we ask ratings for 50 randomly sampled articles (note that these articles are different from the ones annotated in the baseline setting, and therefore, baseline model results may differ). Table xx outlines the results for the unguided setting. The results show that the HYDRASUM model performs on par with the baseline model along all quality dimensions measured, even outperforming it in terms of factuality for both NEWSROOM and XSUM. This agrees with our results from 1 which similarly shows that both the baseline and HYDRASUM model summaries have similar quality.

## G    RELATED WORK - EXTENDED

Prior work in controllable text generation can be broadly divided into two categories: 'content' control and 'style' control. The former body of work aims to influence the content-selection of the

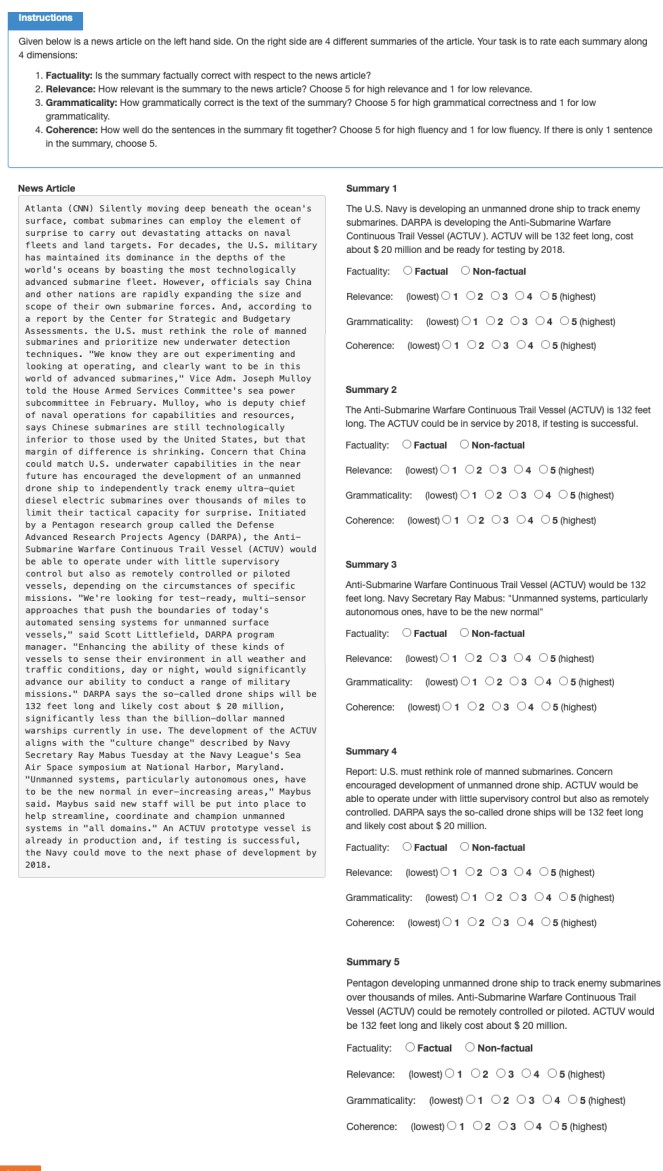

Figure 10: Interface of the Mechanical Turk Task

summarization model, using control variables like keywords (He et al., 2020), queries (Abacha et al., 2021), or even explicit fine-grained content plans e.g. entity-chains (Narayan et al., 2021; Elsahar et al., 2021). These approaches primarily provide 'content-plans' as additional textual inputs to the model, and hence there task formulation differs significantly from our paper. Instead, we focus on a different subset of surface-level summary properties, like length, specificity, readability, etc. that do not target content selection per se. Recently, GeDi (Krause et al., 2021) proposed using small LMs as generative discriminators for specific attributes (e.g. toxicity) to guide the generation of larger models. Similar class-conditional language models approaches (CC-LMs) have been previously proposed (Keskar et al., 2019; Ficler & Goldberg, 2017) to fine-tune models on specific attributes. HYDRASUM models can disentangle styles within the task-specific dataset without explicit style annotations (unguided setting), as well as cover the generation space between two 'extreme' styles (e.g. intermediate abstractiveness level by controlling gate values).

Prior work on 'style' control of text summarization models is discussed in the main body of the paper (see 4). These usually focus on attributes like length (Fan et al., 2018a; Song et al., 2021),

abstractiveness (Song et al., 2020), etc. However, approaches proposed in these works are over-specialized towards the target style and cannot be generalized to more control attributes, or adapted to the multi-control setting. Our HYDRASUM approach, on the other hand, is easily generalizable.

