# OpenReview forum: "HydraSum - Disentangling Stylistic Features in Text Summarization using Multi-Decoder Models"
_ICLR.cc/2022/Conference — ICLR 2022 Submitted_

### Official Review · Reviewer_aQkL · 2021-11-01

**Correctness:** 4
**Technical Novelty And Significance:** 3
**Empirical Novelty And Significance:** 2
**Recommendation:** 6
**Confidence:** 4

**Main Review:**

Strengths:
1. This provides an alternative to controlled generation. Standard solutions so far include control tokens (not cited, see eg [1]) or a-posteriori filtering (cited) which have drawbacks (less flexibility for the former, very expensive for the latter).
2. The idea is simple (Methodology section takes just over 1 page)
3. The paper has an in-depth analysis part, showing convincingly that stylistic features are indeed captured and can be used for control. Most importantly (I consider this the main novelty of this approach), they can be combined. Unfortunately, there is not much space left for detailed analysis on this generalization capacity of the proposed approach.

Weaknesses.
1. The proposed solution only acts on the single token level, as it intervenes by combining experts that predict individual tokens. This is an inherent weakness of auto-regressive models, although recent alternatives have been proposed (see [2] and references therein)
2. The compared baselines do not include any of the existing approaches (like those mentioned in Strength1). Comparing one controlled model against other non-controllable models and not considering how the problem was tackled so far is a pity. The compared models are (probably) much smaller as they do not incur in the extra cost of the additional adapter layers. For fair comparisons, the model sizes should be equivalent


As a potential upper bound for ROUGE (page 6), instead of picking topK, one possibility would be to select for each document the decoder which obtains the best summary.

[1] Self-Supervised and Controlled Multi-Document Opinion Summarization. EACL 2021
[2] A Step-Wise Weighting Approach for Controllable Text Generation. ICLR 2022 submission (K8HF8tTQ-4i)

**Summary Of The Paper:**

This paper proposes an architecture for controlled summarization, allowing some type of aspect-based summarization, as in - for example - "I want long/detailed summary". The control proposed here is on what the authors call _stylistic_ features (which also includes _quality_, arguably not a stylistic aspect) and not other aspect (as in for example: give a summary of Darius the Great focusing on his earlier years).

They obtain this by using a Transformer (BART-like) seq2seq model, but where the last layers of the decoder are multiplied and considered as independent expert which are combined in the end. That combination consists in a weighted average (no non-linearity) of the last softmax layers.

The proposed solution is simple (good!) although not very elegant. The analysis focuses on evaluating the control capacity and diversity with respect to other (smaller) architectures.

**Summary Of The Review:**

A valuable contribution to controlled generation. The evaluation focuses on the strong point of the models (more diversity) and is somehow unfair (against non-controllable models that are smaller). In this sense it is not clear if it is better or just different than existing methods.

---

> ### Author Response · Authors · 2021-11-19
> **Response to R4**
>
> Thank you for your feedback! We’ve addressed the main concerns below:
>
> __1. Token-level vs summary-level:__ Although our gating mechanism is defined at the token-level, the formulation is flexible and allows users to set sentence-level and summary-level labels. Similarly, although energy-based models can evaluate the style/property of generated text at the sentence/phrasal level, their generation aspect still relies on token-level sampling and suffers from similar limitations.
>
> __2. Comparison with other controllable models:__ We want to draw a distinction between prompt/keyword-based ‘content’ control models (cited in our work as well as the citations shared here) and our work that does not aim to control the ‘content’ but instead stylistic-properties like abstractiveness and specificity. Note that our selection of control-properties was informed by the results in the unguided setting (Table 1); we chose properties that exhibited the highest style-partitioning.
>
> ‘Content plan’ based models, e.g. the opinion summarization paper [1] (thanks for the cite!) relies on domain-specific controls tokens (here, they use entity-types like Beauty&Spa, FurnitureStores, etc.) to specify the content plan. However, it requires significant work to hand-design such appropriate control tokens for the news domain, and is out of scope for our work.
>
> Perhaps a more closely related line of work are recent models [3,4] that also aim to control generation by modifying output probabilities. However, these work on a separate set of properties (like toxicity, sentiment for [3] and poetry, topic, and formality for [4]), and do not serve as baselines for our task. Moreover, the assumptions, training data requirements and label requirements for these approaches are not trivially satisfied by the news summarization datasets used in our work. Therefore, adapting their approach for our selection of style-properties is non-trivial and merits a separate paper/discussion.
>
> __3. HydraSum models have more parameters than baseline models:__ We agree that our proposed models are bigger. We attempt to reduce the number of additional parameters by sharing parameters in the lower layers of the multiple decoders (8 out of 12 layers have shared parameters). We did not reduce the hidden state size, etc. of the decoders since we wanted to initialize the model using learned weights from pre-trained models like BART.
>
>
> [1] Self-Supervised and Controlled Multi-Document Opinion Summarization. EACL 2021
> [2] Li et al. AAAI 2015, Fast and Accurate Prediction of Sentence Specificity
> [3] Liu et al. ACL 2021, DExperts: Decoding-time controlled text generation with experts and anti-experts.
> [4] Yang et al. ACL 2021, FUDGE: Controlled text generation with future discriminators.

---

> > ### Comment · Reviewer_aQkL · 2021-11-29
> > **Thanks for the reply**
> >
> > Thank you for your thoughtful answers to me and the other reviewers, answers which I read with attention.
> > On (1) I agree that the connection of sequence-level control is very high-level, and that comparing to it is probably out of scope.
> > I am less convinced of the other two points: many of the things you propose could be done with simple control token (which do not have to be domain specific). Your model (which I like) is strictly more powerful, but that delta is not really put forward in the paper (mainly in the paragraph titled "MULTI-FEATURE CONTROL" on page 8).
> >
> > I hope that - regardless of acceptance of this paper - you have the possibility to continue working on this topic. It seems a highly flexible way of enriching existing models in a simple way for additional control behaviour.

---

### Official Review · Reviewer_kxHp · 2021-11-01

**Correctness:** 3
**Technical Novelty And Significance:** 3
**Empirical Novelty And Significance:** 3
**Recommendation:** 6
**Confidence:** 4

**Main Review:**

Strengths:
- the paper is well motivated and tackles a hard and long-standing problem with seq2seq models: diversity and controllability.
- the approach is fairly simple, interesting and to the best of my knowledge, novel.
- the paper is well structured, clearly explained and free of orthographic/grammatical issues.

Weaknesses:
- It is surprising to use beam search as the only baseline algorithm for generating >1 summaries as it is a poor way to generate diverse outputs. It would have been more interesting to compare with diverse beam search, nucleus sampling, etc, possibly in addition to beam search. Can these algorithms be added to the analysis? Even if metric increases are more modest, the proposed approach still provides better controllability which is interesting in its own right.
- Can the guided setting allow the model to manage gating? Otherwise, this makes the guided setting dependent on manual inputs while not letting the model learn dataset-specific gating. Given the authors used heuristics to partition the datasets, could these partitions be used as a supervision signal for gating? If not, why not?
- The proposed approach has a non-negligible computational cost, compared to using a single decoder, that it would have been interesting to discuss.
- The literature review is fairly short and to the point. Diversity and controllability have been large research areas in the past few years and it would have been helpful to, at the very least, provide more details on the cited papers and contrast them with the proposed approach.

Nits:
- In Figure 1, the baseline summaries are likely generated using beam search. It would be good to mention it explicitly.
- In Table 2, it would be clearer to mention R1/R2/RL as was done in Table 1.
- In Table 2, R2 for XSum is lower than that of the baseline, so it should probably not be written in bold.

**Summary Of The Paper:**

This paper proposes a new architecture for controllable text summarization, leveraging multiple decoders (with the bottom decoder layers  shared across decoders) with the contributions of each decoder controlled by a gating mechanism. This model is studied in two gating settings: unguided (where the network learns the gating weights) and guided (where gating is controlled manually). The networks are shown to allow some stylistic controllability, most notably in terms of abstractiveness and specificity.


**Summary Of The Review:**

Overall this is an interesting paper proposing a new approach to stylistic controllability for neural text summarization. To the best of my knowledge, this approach is novel. I believe this work is interesting as is, but were surprised by some of the experimental settings that I would like to see clarified by the authors.

---

> ### Author Response · Authors · 2021-11-19
> **Response to R3**
>
> Thank you for your reviews and suggestions! We address the issues raised below:
>
> __1. Other decoding strategies for the baseline model:__ We’ve now added results for the baseline model using 2 other decoding schemes, top-k decoding and diverse beam search (see Table 2 in revised draft). Results show that our HydraSum model outperforms all decoding strategies in terms of quality (measured by TopK-Rouge).
>
> In terms of stylistic diversity, we observed that the baseline model + top-k decoding leads to better style diversity. However, this is not accompanied by a corresponding increase in quality (the TopK Rouge is ~2 points lower than HydraSum), which strongly indicates that this additional diversity is achieved by sampling low quality summaries.
>
> Due to time and computational constraints, we could not run experiments using top-k decoding for our HydraSum models. Prior work [1] has shown that it leads to higher diversity than beam search, and we expect to see equivalent gains when using it with our models. We will include these results in a future version.
>
> __2. Training the gating mechanism using heuristics:__  This is an interesting suggestion and definitely merits further investigation, but we don’t feel there is room in the paper to dive into this topic in full detail.
>
> In our initial explorations, we did perform experiments where the token-level oracle gates (extrapolated from the sentence- or summary-level gates used in the paper) were used to provide explicit supervision to the gating mechanism in the guided setting. This was done by adding an additional loss objective for the binary classification task (computed using the predicted token-level softmax values and the oracle gate labels).
>
> Our preliminary results showed that the guided approach (in the paper) led to better style-separation between decoders and due to space constraints, we decided not to include other experiments in the paper.
>
> Moreover, we want to point out that the dataset-specific gating is already learnt by our unguided models, which can be used to generate generic dataset summaries. The guided approach improves over that by enforcing higher style-diversity between individual decoders, and therefore helps cover a larger area in the generation space.
>
>
> __3. Computation cost:__ For the proposed  HydraSum models, decoding cost increases linearly with the number of additional decoders. However, this computation can be parallelized at both training and inference times.
>
>
> __4. Literature review:__ Thanks for the suggestion. We've included a more extensive literature review to clearly distinguish our work from prior approaches. Due to space constraints, it's currently in the Appendix. We will restructure the paper to include it in the main body of the paper in the next version.
>
>
> [1] Ippolito et al. ACL 2019, Comparison of Diverse Decoding Methods from Conditional Language Models

---

### Official Review · Reviewer_5qbJ · 2021-11-02

**Correctness:** 4
**Technical Novelty And Significance:** 3
**Empirical Novelty And Significance:** 3
**Recommendation:** 6
**Confidence:** 4

**Main Review:**

**Novelty**

The fundamental idea of the paper to use multiple decoders and a gating mechanism to improve abstractiveness in model generated summaries is previously explored in [1]. However, the "guided" training scheme is interesting, specially the way the training data is partitioned based on different features.

[1] Improving Abstraction in Text Summarization, EMNLP 2018.

**Strengths**

- Good writing. I enjoyed reading the paper.
- Thorough experiments; the paper covered a broad range of aspects to evaluate the proposed model.
- Overall the findings of the paper is interesting, and it will help push future works in this direction.

**Weaknessess**

- Limited technical novelty.
- As the Table 1 suggests, the overall performance of the proposed model does not surpass the baseline in 2 out of 3 evaluation dataset. This raises a key question, in what scenarios, the proposed model would be valuable? Are there any particular use cases of this model? I do not see any discussion on that.

**Questions**

- Why 2-gram overlapping is used as a measure for abstractiveness? What is the motivation?
- If the feature refers to specificity (as in section 3.2), what is the criteria to split the training dataset?
- Is it possible to perform human evaluation to judge the diversity of the generated summaries?


**Summary Of The Paper:**

This paper proposes a neural sequence-to-sequence (Seq2Seq) model, called HYDRASUM for text summarization. HYDRASUM incorporates multiple decoders where each decoder learns and generates stylistically distinct summaries along dimensions such as abstractiveness, specificity, and others. HYDRASUM is fundamentally built upon the notion of mixture-of-experts (MoE), where a gating mechanism decides the contribution of each expert (individual decoder) to the next token's output probability distribution. The paper demonstrates that HYDRASUM automatically learns to generate contrasting summary styles using each decoder. The paper further proposes a "guided" training scheme that explicitly govern which summary style is partitioned between decoders, e.g. high abstractiveness vs. low abstractiveness or high specificity vs. low specificity, and also increase the stylistic differences between individual decoders. The paper also show that HYDRASUM is flexible, during inference, it can be controlled to generate a diverse set of summaries by sampling from individual decoders or mixtures of different subsets of the decoders.

**Summary Of The Review:**

The novelty of the proposed method is thin. However, the paper presents rigorous experiments and evaluation to validate the main claim. Performing human evaluation to judge the quality of the generated summaries would be beneficial.

---

> ### Author Response · Authors · 2021-11-19
> **Response to R2**
>
> Thank you for your feedback! We’ve addressed the main concerns below:
>
> 1. __Multiple decoder idea already explored in “Improving Abstraction in Text Summarization, EMNLP 2018” paper__: Thanks for the citation! We agree that the idea of using multiple decoders has been explored previously. However, prior work required substantial changes in the models architecture and/or training algorithms, e.g. policy gradients, additional loss penalties, separately trained language model, etc to make it work. On the other hand, we propose a simple modification that introduces no significant changes to the standard architecture and does not modify the loss function at all. This, in turn, helps us benefit from  pre-trained model initializations.
> Also note that we outperform prior work both in terms of quality (this is to be expected because we use transformers + pre-trained models) and level of abstraction achieved, which is significantly higher for our model. Moreover, we provide a way to vary this level of abstraction by using user-specified gate values, while prior models aren’t customizable. Finally, our technique is broadly applicable and can be easily extended to other styles/properties.
>
> 2. __Motivation/ use-cases for the model__: It has been previously shown that the expectations with respect to the features and content of document summaries substantially vary between end users [6]. To meet their needs, summarization systems should allow the users to specify what their preferences are and use that information to guide the generation process. While traditional text summarization systems do not have mechanisms which would allow for summary customization, research in controllable and personalized summarization focuses on such features. In our work we show that while HydraSum is on par or better than generic summarization models with respect to summary quality, it also allows the users to customize the stylistic features of the generated text, thus better catering to their needs. The motivation for our work is further discussed in the Introduction of the paper.
>
> 3. __Why is 2-gram overlap used to measure abstractiveness?__  The use of overlap-driven metrics to measure abstractiveness, particularly extractive fragment coverage, extractive fragment density, repeated n-gram (called n-gram overlap in our paper) is well established in prior text summarization research [1, 2]. We report all these metrics in Table 1. Thereafter, for simplicity, we only report 2-gram overlap as (a) It is bounded between [0,1] while extractive fragment density metric is not, (b) These overlap metrics are mutually-correlated. Table 1 shows that the relative trends for both coverage and density are similar to 2gram overlap across all models and inference strategies.
>
> 4. __If the feature refers to specificity (as in section 3.2), what is the criteria to split the training dataset?__ The strategy used to split the training data is outlined in Section 3.2. While we describe our methodology using f=abstractiveness as an example, we follow the exact same strategy for specificity.
>
> Note though, that specificity is defined at the sentence level. Therefore, we obtain  percentile-splits according to sentence-level specificity scores, and set oracle gates g at the sentence-level.
>
> 5. __Human evaluation for diversity:__ Text generation literature [3, 4] has discussed the difficulty of asking human annotators to rate diversity.  Therefore, we follow prior work in diversity evaluation [5] and rely on automated metrics for evaluation and have now added human evaluation of quality in the revised paper.
>
>
> [1] Grusky et al. NAACL 2018, NEWSROOM: A Dataset of 1.3 Million Summaries with Diverse Extractive Strategies
> [2] Fabbri et al. TACL 2021, SummEval: Re-evaluating Summarization Evaluation
> [3] Hashimoto et al. NAACL 2020, Unifying Human and Statistical Evaluation for Natural Language Generation
> [4]  Montahaei et al. NeuralGen Workshop, NAACL2019, Jointly Measuring Diversity and Quality in Text Generation Models
> [5] Vijaykumar et al. AAAI 2018, Diverse Beam Search for Improved Description of Complex Scenes
> [6] Kryściński et al. EMNLP 2019, Neural Text Summarization: A Critical Evaluation

---

### Official Review · Reviewer_3AAA · 2021-11-03

**Correctness:** 3
**Technical Novelty And Significance:** 2
**Empirical Novelty And Significance:** 2
**Recommendation:** 6
**Confidence:** 4

**Main Review:**

==== strengths ====

The task of controllable generation for summarization is well motivated. The finding that different decoders indeed capture different level of abstractiveness and specificity is interesting. The paper is clearly written and the methods are experiments are easy to understand.

==== weaknesses ====

My biggest concern with this work is lacking of good evaluation metrics. For the same text input, the model is supposed to control all high-quality summaries with distinct styles. To achieve that, a summarization dataset with multiple stylistically distinct references are needed. Lacking such a dataset, human evaluation would shine some light, which is missing in this work. In the guided training session, what is the rouge score of generated summaries?

**Summary Of The Paper:**

This paper propose using a multidecoder architecture to to control the style of output summarization. In the designed model, each decoder learns and generates stylisticall-distinct summaries. The contribution of each experts are controlled by a gating mechanism. This multidecoder model can be trained without supervision or with guidance. This model is tested on CNN, NEWSROOM and XSum. When trained unsupervised, different decoders learn to produce summaries with different abstractiveness and specificity. The guided training shows the capability of asigning certain styles to a specific encoder.

**Summary Of The Review:**

The paper presented a novel architecture for style control in abstractive summarization. However, without evaluating the model on a dataset where each input is associated with multiple references of different styles or running human evaluation, the effectiveness of the style control and the quality of generated summaries are not clear.

---

> ### Author Response · Authors · 2021-11-19
> **Response to R1**
>
> Thank you for your constructive feedback! We address the issues raised below:
>
> __Lack of human evaluation__: As mentioned in the general response, we’ve included human evaluation of summary quality. Quantitative experiments in Section 3.1 and Section 3.2 already showed that summaries generated using D0 and D1 are stylistically-distinct from each other. Through our human evaluation, we show that this increased stylistic diversity is achieved while maintaining high generation quality, on par with baseline models.
>
> __Rouge scores for the guided setting__: Rouge scores, Topk-Rouge scores and other metrics for the guided setting have been included in Appendix D, Table 10 of the original submission.

---

### Author Response · Authors · 2021-11-19
**General response to all reviewers**

We thank all the reviewers for their valuable comments! Based on the feedback, we have revised the paper making the following major change:

__Added human evaluation of summary quality (R1, R2)__: We’ve included human evaluation to compare summaries generated by HydraSum and baseline models, for both the guided (Table 5) and the unguided (Table 11) settings. For the same input article, we asked Mechanical Turk workers to rate generated summaries from different models along 4 quality metrics: relevance, coherence, grammaticality, and factuality. Details are included in Section 3.2 (Human Evaluation) and Appendix F.
Our results show that the HydraSum model outperforms (for the guided setting, Table 5) or is on par (for the unguided setting, Table 11) with baseline models along these different dimensions. This shows that the proposed approach does not sacrifice summary quality while exposing a style-controllable lever of text summarization models.

We address other reviewer concerns in individual responses.

---

### Decision · Program_Chairs · 2022-01-20

**Decision:**

Reject

**Comment:**

The paper is well motivated and tackles a hard and long-standing problem with seq2seq models: diversity and controllability.
The authors propose simple architecture for controllable text summarization. They use multiple decoders controlled by a gating mechanism which can be learnt or controlled manually. They control mostly the abstractiveness and specificity properties of the model.

Pros
+ the proposed approach is somewhat novel (several earlier work have proposed multiple decoder models to control the generation -- as pointed by the reviewer team)
+ the proposed modifications are motivated well, the approach is simple and easy to understand.
+ the paper is well written and easy to read.
+ the authors made an effort to address most of the reviewers comments even added human evaluation scores (which was asked by reviewers)
+ It seems a highly flexible way of enriching existing models in a simple way for additional control behavior in output summary generation of documents.

Cons
+ During discussions, reviewers have circled around the novelty and continued to raise concerns about the weaknesses of benchmarks and comparison to related work and the fact that the proposed model has more parameters is potential advantage over other models that might contribute to the performance gains. Thus, the paper could be made stronger with further evaluations that could possibly make it stand out.